# Prevalence of uncoupling protein one genetic polymorphisms and their relationship with cardiovascular and metabolic health

Petros C. Dinas[1,2☯], Eleni Nintou[1☯], Maria Vliora[1☯], Anna E. Pravednikova[3,4], Paraskevi Sakellariou[1], Agata Witkowicz[5], Zaur M. Kachaev[3], Victor V. Kerchev[3,4], Svetlana N. Larina[3,4], James Cotton[6], Anna Kowalska[7], Paraskevi Gkiata[1], Alexandra Bargiota[8], Zaruhi A. Khachatryan[9], Anahit A. Hovhannisyan[9], Mariya A. Antonosyan[9], Sona Margaryan[9], Anna Partyka[5], Pawel Bogdanski[10], Monika Szulinska[10], Matylda Kregielska-Narozna[10], Rafał Czepczyński[11], Marek Ruchała[11], Anna Tomkiewicz[5], Levon Yepiskoposyan[12], Lidia Karabon[5], Yulii Shidlovskii[3,4], George S. Metsios[13], Andreas D. Flouris[1]*

**1** FAME Laboratory, Department of Physical Education and Sport Science, University of Thessaly, Trikala, Greece, **2** Faculty of Education Health and Wellbeing, University of Wolverhampton, Walsall, West Midlands, United Kingdom, **3** Laboratory of Gene Expression Regulation in Development, Institute of Gene Biology, Russian Academy of Sciences, Moscow, Russia, **4** Department of Biology and General Genetics, Sechenov First Moscow State Medical University (Sechenov University), Moscow, Russia, **5** L. Hirszfeld Institute of Immunology and Experimental Therapy, Polish Academy of Sciences, Wrocław, Poland, **6** Royal Wolverhampton NHS Trust, New Cross Hospital, Wolverhampton, United Kingdom, **7** Institute of Human Genetics, Polish Academy of Sciences, Poznań, Poland, **8** Department of Endocrinology and Metabolic Diseases, Medical School, Larissa University Hospital, University of Thessaly, Larissa, Greece, **9** Institute of Molecular Biology, National Academy of Sciences of the Republic of Armenia, Yerevan, Armenia, **10** Department of Treatment of Obesity, Metabolic Disorders and Clinical Dietetics, Poznan University of Medical Sciences, Poznań, Poland, **11** Department of Endocrinology, Metabolism and Internal Medicine, Poznan University of Medical Sciences, Poznań, Poland, **12** Department of Bioengineering, Bioinformatics and Molecular Biology, Russian-Armenian University, Yerevan, Armenia, **13** Department of Nutrition and Dietetics, School of Physical Education, Sport Science and Dietetics, University of Thessaly, Trikala, Greece

☯ These authors contributed equally to this work.

* andreasflouris@gmail.com

**Data Availability Statement:** Datasets related to this study have been uploaded on Figshare (https://figshare.com/articles/dataset/UGENE_project_

## Abstract

Contribution of *UCP1* single nucleotide polymorphisms (SNPs) to susceptibility for cardio-metabolic pathologies (CMP) and their involvement in specific risk factors for these conditions varies across populations. We tested whether *UCP1* SNPs A-3826G, A-1766G, Ala64Thr and A-112C are associated with common CMP and their risk factors across Armenia, Greece, Poland, Russia and United Kingdom. This case-control study included geno-typing of these SNPs, from 2,283 Caucasians. Results were extended via systematic review and meta-analysis. In Armenia, GA genotype and A allele of Ala64Thr displayed ~2-fold higher risk for CMP compared to GG genotype and G allele, respectively ($p<0.05$). In Greece, A allele of Ala64Thr decreased risk of CMP by 39%. Healthy individuals with A-3826G GG genotype and carriers of mutant allele of A-112C and Ala64Thr had higher body mass index compared to those carrying other alleles. In healthy Polish, higher waist-to-hip ratio (WHR) was observed in heterozygotes A-3826G compared to AA homozygotes. Het-erozygosity of A-112C and Ala64Thr SNPs was related to lower WHR in CMP individuals compared to wild type homozygotes ($p<0.05$). Meta-analysis showed no statistically

DATABASE_All_participants_xlsx/17206709)
under doi: 10.6084/m9.figshare.17206709.

**Funding:** • European Union 7th Framework
Program (FP7-PEOPLE-2013-IRSES Grant No.
319010; U-GENE project • Russian Science
Foundation grant 20-14-00201 (case-control study
in the Russian population; meta-analysis). • Polish
Ministry of Science and Higher Education 2016-
2017 international project co-financed W15/7.PR/
2016. The funders had no role in study design, data
collection and analysis, decision to publish, or
preparation of the manuscript.

**Competing interests:** The authors have declared
that no competing interests exist.

significant odds-ratios across our SNPs (p>0.05). Concluding, the studied SNPs could be associated with the most common CMP and their risk factors in some populations.

## Introduction

Single nucleotide polymorphisms (SNPs) in a number of candidate genes are highly implicated in energy balance as well as fat and glucose metabolism, modifying disease susceptibility [1–3]. One of these candidate genes codes for uncoupling protein 1 (UCP1), located on chromosome 4 (4q31.1), which is expressed predominantly in brown adipose tissue, holding a critical role in oxidative phosphorylation and overall energy balance [4, 5]. More than 2300 SNPs have been recognized within the *UCP1* gene and its regulatory regions [6], but four have been commonly studied for their impact on metabolism and energy balance [7–11]. These are: (i) A-3826G (rs1800592) located on the upstream region of *UCP1*, (ii) A-1766G (rs3811791) a 2kb upstream variant, (iii) A-112C (rs10011540) on the 5'UTR region, and (iv) Ala64Thr (rs45539933) a missense variant.

The four *UCP1* SNPs have been associated with a number of cardio-metabolic pathologies (CMP) [12]. The G allele of A-3826G, which is associated with reduced mRNA expression of *UCP1* [13], is more common in obese individuals [14, 15] and it is associated with increased body mass index (BMI), percent body fat, blood pressure [16], and lower high-density lipoprotein level [17]. The same allele of this SNP is associated with higher BMI and glucose levels in overweight persons [18] and can increase the risk for proliferative diabetic retinopathy in individuals with type 2 diabetes [19]. The other three SNPs are less prevalent but have been also associated with various risk factors for CMP [6, 11]. The A-112C polymorphism affects *UCP1* gene promoter activity [20] and the C allele is more frequent in individuals with type 2 diabetes than in healthy individuals [21]. The Ala64Thr mutant allele is associated with higher waist-to-hip ratio (WHR) [22], while the A-1766G SNP, which is detected in the genomic region that possibly regulates transcription of *UCP1* [23], is related with obesity [7]. Finally, the GAA haplotype (A-3826G, A-1766G, and Ala64Thr) is associated with decreased abdominal fat tissue, body fat mass, and WHR [24].

The contribution of the four *UCP1* SNPs to the susceptibility for CMP as well as their involvement in specific risk factors for these conditions varies across populations, even within the same race, probably due to environmental impacts. For instance, the frequency of AG genotype of A-3826G in persons with CMP ranges from 24% in Italy [25], to around 50% in Colombia [20], Japan [21], and Korea [17], and to 85% in China [19]. Similarly, wide frequency ranges have been reported also for the other three SNPs across different populations [10, 20, 26, 27]. At the same time, some studies report that *UCP1* SNPs are strongly associated with disease risk [7, 19, 28], while others report no such findings [29–31]. Therefore, it remains unclear if differences in the prevalence of these four *UCP1* SNPs across different populations are associated with the prevalence of CMP.

Our incomplete understanding about the potential involvement of these four *UCP1* SNPs, among others, in disease susceptibility limits the potential for precision medicine to effectively address CMP. An even more direct effect on disease mitigation is that CMP risk factors are currently addressed with equal importance across different populations, ignoring the genotypic/phenotypic complexity of CMP in different countries. Improving our knowledge about the impact of UCP1 variants can contribute to precision medicine, within the context of approaches that consider the polygenicity of cardio-metabolic traits (e.g., polygenic risk

scores). This could improve the sustainability of healthcare systems due to increased efficacy of CMP prevention and mitigation guidelines. To address these important knowledge gaps, we investigated if differences in the frequency of A-3826G, A-1766G, Ala64Thr and A-112C SNPs are associated with the most common CMP and their risk factors. This case control study was performed across five countries (Armenia, Greece, Poland, Russia, United Kingdom) since CMP appear to be increased in certain ethnic groups in Eastern Europe and Western Asia [32, 33].To confirm any observed associations between the studied *UCP1* SNPs and cardio-metabolic health, we extended our findings to consider all previously-studied populations by conducting a systematic review and meta-analysis [34]. The literature includes four meta-analyses [29, 35–37] regarding *UCP1* SNPs and their association with cardio-metabolic traits. Within these four meta-analyses only A-3826G is examined for its association with metabolic diseases or their risk factors, as the most common variant of *UCP1*, while these meta-analyses do not consider the associations of other *UCP1* SNPs with the risk for disease.

## Materials and methods

### Case-control study

This is a multicenter, multinational study conducted during 2016–2019, across five countries (Armenia, Greece, Poland, Russia, and United Kingdom). The participants were recruited via online and paper advertisements as well as word of mouth. Following approval from the relevant Bioethics Review Board in each country (see Section 1.1.1 in S1 File). Written informed consent for participation was signed by the volunteers following detailed explanation of all the procedures and risks involved.

### Study design and data collection

The study involved two groups of participants: individuals with CMP as well as healthy controls. We considered the following CMP, as they present with the highest prevalence [38, 39] amongst all health abnormalities related to cardio-metabolic health: cardiovascular disease, hypertension, metabolic syndrome, and type 2 diabetes. The inclusion criteria were: 1) adult; 2) diagnosed presence of CMP for the CMP group and generally healthy (free of CMP based on their medical history) for the control group; 3) non-smokers, or have quit smoking for at least one year; 4) not in a pregnancy or lactation period; 5) no history of eating disorders; 6) no acute illness and/or infection during the last four weeks.

Ethnicity was self-reported by each participant. All participants were assessed for: 1) medical history via a structured interview-based questionnaire; 2) anthropometry (body height, body mass, WHR); 3) percent fat mass via non-invasive bioelectrical impedance analysis; 4) genotypes of the aforementioned four *UCP1* SNPs detected in DNA isolated from blood samples. A detailed description of the adopted blood handling and genotyping methodologies is provided in Section 1.1.2 in S1 File. All participants were instructed, for 12 hours prior to assessments, to avoid the consumption of food, coffee, or alcohol and to refrain from exercise. Also, they were advised to consume two glasses of water about two hours prior to their assessment.

### Statistical analysis

The data were analyzed using a general genetic model as previously described [40, 41]. We calculated Hardy-Weinberg equilibrium to ensure unbiased outcomes [42]. Linkage disequilibrium between genetic loci, haplotype analysis, and allele frequencies estimation were performed via the SHEsis platform [43, 44]. We used chi-square tests to determine differences

in *UCP1* SNPs between groups, as well as Phi indices to report effect sizes [45]. Also, we calculated odds ratios (OR) to determine associations of genotypes and alleles between groups in the overall sample as well as based on country (Section 1.1.3 in S1 File). Finally, we used Kruskal Wallis ANOVA with post hoc Mann-Whitney U tests to assess differences in BMI, WHR, and fat percentage between genotype groups for each *UCP1* SNP. The level of statistical significance for the Hardy-Weinberg equilibrium was set at $p<0.05$ and for all other analyses at $p\leq0.05$. We did not adjust for multiple comparisons in our study due to the errors and misplaced emphasis associated with such procedures when applied in actual natural observations [46–49]. Unless stated otherwise, the SPSS 26.0 (SPSS Inc., Chicago, IL, USA) software was used to perform the statistical analyses.

## Systematic review and meta-analysis

We conducted a systematic review and meta-analysis (PROSPERO review protocol: CRD42019132376) investigating if differences in the frequency of A-3826G, A-1766G, Ala64Thr and A-112C SNPs are associated with the prevalence of the studied CMP. Following the Preferred Reporting Items for Systematic Reviews and Meta-analyses (PRISMA) guidelines [50], we searched the titles and abstracts in PubMed central, Embase, and Cochrane Library (trials) databases from the date of their inception to February 23, 2021, for studies that evaluated the prevalence of *UCP1* A-3826G, A-1766G, Ala64Thr and A-112C SNPs and their association with CMP. No date, participants' health status, language, or study design limits were applied. A detailed description of the systematic review methodology and the searching algorithm is provided in Section 2.1 in S1 File.

## Results

### Case-control study

**Associations between genotype frequencies and health status.** The study population included 2283 Caucasian individuals (Table 1). Our Hardy-Weinberg equilibrium (HWE) analysis for the A-1766G revealed significant deviation in healthy individuals ($\chi2 = 33.34$, $p<0.001$), indicating that this SNP should be excluded from further analysis [42], for other

**Table 1. Characteristics of the studied population.**

|  | Group | (n) / (%) | Males / Females (n) | Age (years) | BMI (kg/m$^2$) |
|---|---|---|---|---|---|
| **Entire sample** | Healthy | 1139 / 50 | 762 / 528 | 45 (32,54) | 25.5 (23.9,26.9) |
|  | CMP | 1144 / 50 | 397 / 521 | 59 (50,65) | 30.5 (27.4,34.2) |
| **Armenia** | Healthy | 105 / 32 | - | - | - |
|  | CMP | 226 / 68 | 98 / 128 | 59 (54,64) | 29.0 (27.2,31.7) |
| **Greece** | Healthy | 233 / 47 | 131 / 102 | 55 (50,65) | 26.8 (24.2,29.9) |
|  | CMP | 264 / 53 | 125 / 139 | 62 (56,68) | 31.7 (28.9,34.5) |
| **Poland** | Healthy | 365 / 59 | 221 /144 | 32 (25,44) | 23.8 (22.0,25.6) |
|  | CMP | 252 / 41 | 89 /163 | 62 (54.7,67) | 31.2 (29.4,33.8) |
| **Russia** | Healthy | 255 / 45 | 142 / 113 | 46 (36.5,54.5) | 25.9 (25.3,26.3) |
|  | CMP | 310 / 55 | 129 / 181 | 52 (40,63) | 28.9 (26.0,34.6) |
| **UK** | Healthy | 181 / 66 | 140 / 41 | 43 (30,51) | 25.7 (23.2,29.8) |
|  | CMP | 92 / 34 | 54 / 38 | 54 (48,57) | 30.7 (25.9,38.4) |

Note: Age and BMI are presented as median (Q1, Q3).

Key: CMP = cardio-metabolic pathologies; BMI = body mass index; n = number of individuals tested. Q = quartile).

*UCP1* SNPs no deviation from HWE in healthy individuals was noticed. The frequencies of alleles and genotypes for the studied *UCP1* SNPs in healthy controls and in CMP individuals are shown in Fig 1, Table 2 and S4–S11 Tables in S1 File. Odds ratios for the association between genotype and health status (i.e., healthy vs. CMP individuals) for each of the four studied *UCP1* SNPs are shown in Table 2 and S10 and S11 Tables in S1 File.

With regard to country-level stratification, allele frequency analysis (S4–S9 Tables in S1 File) in the Greek population showed that individuals carrying the C allele of the A-112C SNP or the A allele of the Ala64Thr SNP are 37% and 39% less likely to develop CMP, respectively (p<0.05; S6 Table in S1 File). Moreover, the G allele of the A-3826G SNP was associated with 23% lower risk to develop CMP in the Polish population (S7 Table in S1 File).

In total, we found no associations between genotype and health status in the overall sample for the studied *UCP1* SNPs (p>0.05). Though, we observed an association between genotype and health status for Ala64Thr within the Armenian population, where the GA genotype was carried by 24.4% of the CMP individuals but only by 13.5% of healthy individuals. Also, the GA genotype of Ala64Thr showed a 2-fold higher risk (p = 0.03) for CMP than the GG genotype in the Armenian population (Table 2).

**Linkage disequilibrium.**   Our analysis for all four SNPs in this study in CMP individuals and healthy controls showed that the A-3826G and Ala64Thr were in strong linkage disequilibrium with a D' value of 0.831. Similar results were observed for the combinations of A-3826G and A-112C, as well as for the Ala64Thr and A-112C which were in strong linkage disequilibrium with D' values of 0.917 and 0.924, respectively. However, the $r^2$ values for the combinations of A-3826G and Ala64Thr ($r^2$ = 0.165) as well as A-3826G and A-112C ($r^2$ = 0.195) were relatively low, indicating that their effects are independent of each other. In contrast, the $r^2$ value for Ala64Thr and A-112C was high ($r^2$ = 0.848), indicating a direct link between these two SNPs. Country-specific analysis of linkage disequilibrium between investigated SNPs can be found in S1 and S2 Figs in S1 File.

**Haplotype analysis.**   In the overall sample, the haplotype analysis revealed that CMP individuals were 24% less likely to carry the GAC (A-3826G, Ala64Thr, A-112C) haplotype compared to healthy controls (OR: 0.76 CI95%: 0.60–0.96 p = 0.023; S1 Table in S1 File). Country-

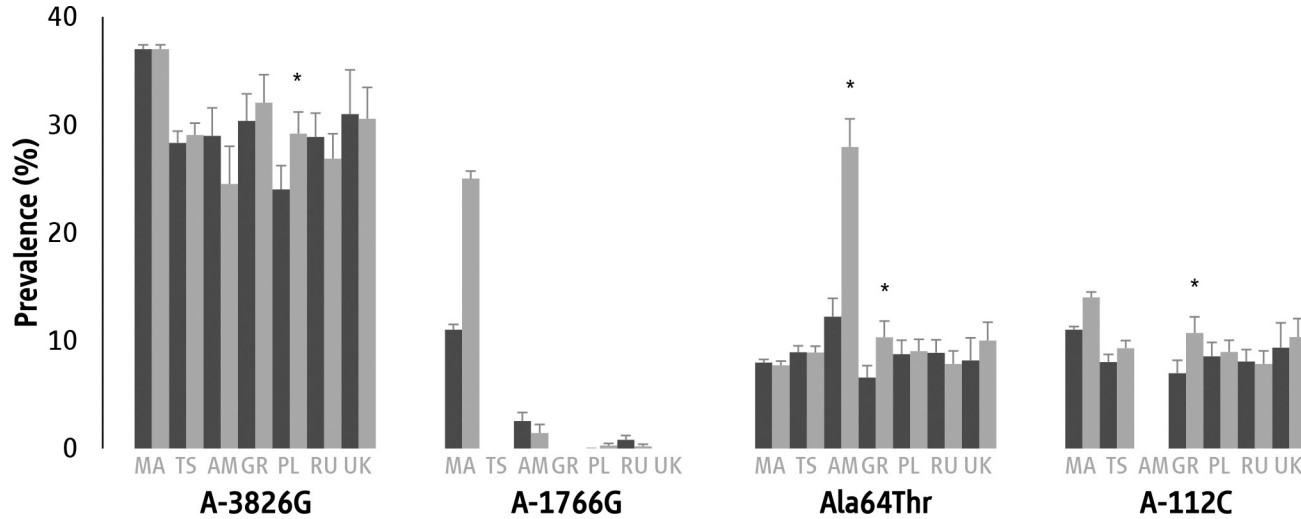

**Fig 1. Prevalence of the studied *UCP1* SNP alleles.** Note: black bars indicate results for individuals with CMP; gray bars indicate results for healthy persons; * indicates differences from CMP persons significant at p<0.05. Key: MA = meta-analysis, TS = total sample, AM = Armenia, GR = Greece, PL = Poland, RU = Russia, UK = United Kingdom.

**Table 2. Frequency of genotypes for Ala64Thr in CMP and healthy individuals.**

| | | Healthy | | CMP | | OR (95% CI) | F-test |
|---|---|---|---|---|---|---|---|
| | | (n) | (%) | (n) | (%) | | |
| **Total sample** | GG | 944 | 83.39 | 928 | 82.71 | | 4.03 p = 0.203 |
| | GA | 175 | 15.46 | 188 | 16.76 | 1.09 (0.87–1.37) | |
| | AA | 13 | 1.15 | 6 | 0.53 | 0.49 (0.19–1.25) | |
| | HWE | 0.134 | | 0.284 | | | |
| **Armenia** | GG | 90 | 86.54 | 164 | 75.58 | | 5.70 p = 0.031 |
| | GA | 14 | 13.46 | 53 | 24.42 | 2.03 (1.08–3.83) | |
| | AA | 0 | 0.00 | 0 | 0.00 | — | |
| | HWE | 0.462 | | 0.040 | | | |
| **Greece** | GG | 184 | 80.70 | 219 | 87.25 | | 4.25 p = 0.115 |
| | GA | 41 | 17.98 | 31 | 12.35 | 0.64 (0.39–1.05) | |
| | AA | 3 | 1.32 | 1 | 0.40 | 0.36 (0.05–2.46) | |
| | HWE | 0.679 | | 0.931 | | | |
| **Poland** | GG | 304 | 83.29 | 211 | 83.73 | | 0.39 p = 0.842 |
| | GA | 58 | 15.89 | 38 | 15.08 | 0.95 (0.61–1.48) | |
| | AA | 3 | 0.82 | 3 | 1.19 | 1.44 (0.32–6.40) | |
| | HWE | 0.899 | | 0.394 | | | |
| **Russia** | GG | 218 | 85.49 | 257 | 82.90 | | 1.52 p = 0.454 |
| | GA | 34 | 13.33 | 51 | 16.45 | 1.27 (0.79–2.02) | |
| | AA | 3 | 1.18 | 2 | 0.65 | 0.61 (0.12–3.10) | |
| | HWE | 0.215 | | 0.758 | | | |
| **UK** | GG | 148 | 82.22 | 77 | 83.70 | | 1.65 p = 0.480 |
| | GA | 28 | 15.56 | 15 | 16.30 | 1.04 (0.53–2.05) | |
| | AA | 4 | 2.22 | 0 | 0.00 | 0.21 (0.01–4.01) | |
| | HWE | 0.069 | | 0.395 | | | |

Key: CMP = cardio-metabolic pathologies; OR = odds ratio; HWE = p value for the Hardy-Weinberg equilibrium.

specific analysis showed lower CMP risk for this haplotype across countries but this association reached statistical significance only in the Greek population (OR = 0.56, CI95%: 0.34–0.91, p = 0.017). Additionally, in the Polish population, we found a higher frequency of the AGA haplotype in CMP individuals compared to healthy persons (74.9% vs 70.6%), which indicates the relationship between this haplotype and higher risk of CMP (OR = 1.33, CI95%: 1.03–1.73, p = 0.032). On the contrary, for GGA haplotype we found a lower frequency in CMP Polish population compared to healthy individuals (15.6% vs 20.3%) indicating a protective effect in healthy individuals (OR = 0.74, CI95%: 0.55–0.99, p = 0.047). In the Armenian population, the AA haplotype (A-3826G, Ala64Thr) increased the CMP risk more than 4-fold (OR = 4.10, CI95%: 1.12–14.98, p = 0.02), while the AG haplotype decreased the susceptibility to CMP (OR = 0.65, CI95% = 0.45–0.95, p = 0.025). The AA haplotype differs from the AG in the second position defined by the mutant allele of Ala64Thr confirming the association of A allele of this SNP with CMP risk. Detailed results for haplotype analysis for each country are provided in S1 and S2 Tables in S1 File.

**Association between UCP1 SNPs with specific CMP risk factors.** In healthy individuals, we observed significantly higher BMI in the homozygotes GG of A-3826G as compared to AA and AG individuals (p = 0.03) as well as in carriers of the mutant allele of A-112C (p = 0.015), and Ala64Thr (p = 0.004) compared to the wild type homozygotes (Table 3). We also showed that CMP individuals being heterozygotes of A-112C and Ala64Thr had lower WHR than wild

**Table 3. Body mass index and waist-to-hip ratio [median (Q1, Q3)] across the different *UCP1* SNPs for the entire sample as well as across healthy controls and individuals with CMP.**

| SNP | Genotype | Body mass index | | Waist-to-hip ratio | |
|---|---|---|---|---|---|
| | | Healthy | CMP | Healthy | CMP |
| **A-3826G** | AA | 25.6 (23.5,26.6) | 30.3 (27.4,34.1)[1] | 0.87 (0.81,0.93) | 0.97 (0.92,1.04)[1] |
| | AG | 25.4 (23.6,27.0) | 30.7 (27.5,34.2)[1] | 0.88 (0.81,0.93) | 1.00 (0.92,1.04)[1] |
| | GG | 26.2 (24.1,28.7)[2,3] | 30.8 (27.2,33.8)[1] | 0.88 (0.80,0.92) | 1.00 (0.92,1.05)[1] |
| **A-112C** | AA | 25.4 (23.5,26.7) | 30.6 (27.5,34.2)[1] | 0.87 (0.81,0.93) | 0.98 (0.93,1.04)[1] |
| | AC | 25.9 (23.7,28.3)[2] | 31.2 (27.3,34.2)[1] | 0.88 (0.82,0.94) | 0.96 (0.87,1.02)[1,2] |
| | CC | 26.3 (25.5,27.2) | 27.9 (27.3,32.5)[1] | 0.87 (0.85,0.89) | 0.94 (0.84,1.00) |
| **Ala64Thr** | GG | 25.4 (23.4,26.7)[2,3] | 30.5 (27.4,34.10)[1] | 0.87 (0.81,0.93) | 0.98 (0.93,1.04)[1,3] |
| | GA | 26.0 (23.8,28.3) | 30.5 (27.3,33.7)[1] | 0.88 (0.82,0.93) | 0.97 (0.87,1.03)[1] |
| | AA | 26.3 (26.1,27.4) | 29.8 (27.2,32.7) | 0.90 (0.84,0.98) | 0.92 (0.80,1.02) |

Note

[1] = difference from healthy significant at p≤0.05

[2] = difference from AA significant at p≤0.05

[3] = difference from AG significant at p≤0.05. Key: CMP = cardio-metabolic pathologies

type homozygotes (Table 3). Country-specific analysis showed that in the healthy Greek population, heterozygous individuals of A-112C and Ala64Thr displayed higher BMI and fat mass compared to the wild type homozygotes (BMI p = 0.005, body fat p = 0.008 and BMI p = 0.002, body fat p = 0.005, respectively; S14 Table in S1 File). In the Polish healthy population, mutant homozygotes of the A-112C SNP presented higher BMI compared to heterozygotes and wild type homozygotes (S12 Table in S1 File; p<0.05). Due to linkage disequilibrium between A-112C and Ala64Thr, the same effect was observed for mutant homozygotes of Ala64Thr. Finally, in Polish healthy individuals, higher WHR was observed in GA heterozygotes (p = 0.03) in comparison to wild type homozygous subjects (S12 Table in S1 File).

## Systematic review and meta-analysis

**Searching procedure.** The searching procedure retrieved 817 publications of which 109 were duplicates. We excluded 219 publications being reviews, editorials, and conference proceeding as well as 161 publications which referred to animal studies. From the 328 remaining publications, 276 were excluded as they did not meet the inclusion criteria. In total, 52 eligible publications were included in the analysis. Detailed searching procedure results can be found in a PRISMA flowchart (S3 Fig in S1 File).

**Characteristics of included studies and risk of bias assessment.** The 52 eligible publications included in the analysis were published between 1998 and 2020 and included data from 24 different countries. The extracted data for all 52 included publications can be found in S17 Table in S1 File. The risk of bias assessment demonstrated low risk for the vast majority of the eligible studies (Section 2.2 in S1 File).

**Meta-analysis outcomes.** Fifty-one out of the 52 eligible publications [7, 8, 10, 12, 13, 16–21, 25–31, 51–83] were used for prevalence meta-analyses, while 22 eligible publications were used for odds ratios meta-analyses. The results from the meta-analyses are summarized in Fig 1 and Table 4, while the SNP-specific forest and funnel plots for the prevalence (S5–S24 and S35–S44 Figs in S1 File) and the odds ratios (S25–S34 and S45–S49 Figs in S1 File) can be found in Sections 2.2.1 and 2.2.2 in S1 File. On the whole, for the different genotypes and alleles we performed 24 prevalence meta-analyses and 12 odds ratios meta-analyses which

**Table 4. Meta-analysis results for the prevalence and odds ratios of genotypes of the four different SNPs, between healthy and CMP individuals.**

| SNP | n | Genotypes | Prevalence meta-analyses | | OR meta-analyses | |
|---|---|---|---|---|---|---|
| | | | Healthy (%) | CMP (%) | OR (95%CI) | p |
| A-3826G | 18568 | AA | 43 | 42 | | |
| | | AG | 43 | 43 | 1.02 (0.96–1.09) | 0.46 |
| | | GG | 14 | 15 | 1.06 (0.96–1.17) | 0.23 |
| A-112C | 6153 | AA | 77 | 78 | | |
| | | AC | 21 | 21 | 1.07 (0.80–1.44) | 0.65 |
| | | CC | 2 | 1 | 0.92 (0.65–1.32) | 0.67 |
| Ala64Thr | 4984 | GG | 85 | 82 | | |
| | | GA | 14 | 17 | 1.07 (0.91–1.27) | 0.41 |
| | | AA | 1 | 1 | 0.64 (0.24–1.67) | 0.36 |
| A-1766G | 4608 | AA | 64 | 66 | | |
| | | AG | 30 | 29 | 1.12 (0.81–1.55) | 0.51 |
| | | GG | 6 | 5 | 1.04 (0.53–2.04) | 0.90 |

Key: CMP = cardio-metabolic pathologies; n = number of studied individuals; OR = odds ratio with reference to AA; 95%CI = 95% confidence intervals; p = p value for the Z test indicating the overall effect in the meta-analysis.

included a total of 34,313 cases. No statistically significant differences were observed in the prevalence of the mutant alleles of the four different SNPs ($p > 0.05$; Fig 1). Also, when we considered only case-control studies, we found no statistically significant odds ratios in different alleles across the four studied SNPs ($p > 0.05$).

## Discussion

Our findings confirm an association between the studied *UCP1* SNPs and cardiometabolic health in a multi-country sample of 2,283 persons. Furthermore, we found that differences in the distribution of genotypes and alleles of the studied SNPs between CMP individuals and healthy controls are associated with the prevalence of one or more of the most common CMP and their risk factors, in some (Armenia, Greece, and Poland) but not all (Russia and United Kingdom) countries.

Within our study population, the A-3826G (AG) was the most prevalent of the four SNPs. In persons with CMP, the prevalence was 40%, ranging from 34% in the UK to 42% in Armenia and Russia. This is very similar to the 43% found in our meta-analysis, and mid-way between the 29% reported in Spain [16] and the ~50% reported in Colombia [8], Japan [21], and Korea [17]. Our findings in the case-control study indicate that the A-3826G is not associated with CMP, but that it leads to increased BMI within the healthy population. Thus, it may promote the development of CMP in the presence of environmental factors [84] as well as other genetic traits [85].

Our results for Ala64Thr and A-112C indicate a strong linkage disequilibrium between the two SNPs. In our study the mutant A allele of Ala64Thr was detected in 9% of both healthy individuals and persons with CMP, and this frequency was not very different across the five studied countries. This was similar to the 7% for healthy and 9% for CMP individuals found in our meta-analysis that included data from 4984 persons across nine countries. Our observed prevalence rates for the C allele of A-112C were 9% in healthy persons and 8% in individuals with CMP. This was somewhat lower than the 12% prevalence found in our meta-analysis that included data from 6,153 persons across eight countries. In terms of health impacts, we showed that the Ala64Thr and A-112C are associated with opposing effects in healthy

individuals and persons with CMP. Our results indicate that the A-112C mutant allele demonstrates its effect when present in its heterozygous form and this may be the reason for C allele's association with decreased risk for CMP development. Specifically, we found that healthy individuals carrying the mutant alleles display higher BMI and, in some countries, body fat percent. On the other hand, persons with CMP who carry the mutant variants have lower WHR. These results partly reflect those reported in previous studies [22, 24]. For instance, the presence of mutant alleles Ala64Thr and A-1766G, in combination with A-3826G, can augment the beneficial effects of caloric restriction resulting in greater reductions in WHR [22]. Unfortunately, we were not able to assess potential associations of these SNPs with biochemical indices or with additional clinical features.

It is important to consider the functional impact of A-3826G, A-1766G and Ala64Thr, which is clear since they directly affect the expression of *UCP1*. In the case of A-112C, it is important to also consider the effect of another variant, rs72941746, that is in linkage disequilibrium [86]. The A-112C seems to modify 4 transcription factor binding sites and its region has specific patterns of chromatin accessibility in several tissues. It appears that the linked variant is responsible for much more alterations in transcription factor binding site motifs and consequently the binding of other proteins. This indicates that the association observed in this study when A-112C is present could possibly be an effect of rs72941746 influence.

Our findings indicate potential limitations of common analysis of different races, ethnicities, and regions when analyzing our data as an entire sample or via meta-analytic methods. For instance, the frequency of A allele of Ala64Thr across all our studied countries was 9%, similar to the 8% found in our meta-analysis, in both cases suggesting no differences between healthy persons and individuals with CMP. However, our country-specific analysis demonstrated that the prevalence of A allele of Ala64Thr was significantly higher in healthy individuals across the Armenian (27.9%) and the Greek (10.3%) populations, as compared to CMP persons. Considering risk factors, we detected a number of associations with the four studied SNPs across Greece, Armenia and Poland, which were not observed in the other countries. Taken together, these findings suggest that the studied SNPs may be important for promoting risk factors and pathophysiological mechanisms involved in CMP, but that this involvement may be stronger in some races, ethnicities, and/or regions. Nevertheless, it is important to also note that the increased CMP prevalence in certain ethnic groups in Eastern Europe and Western Asia [32, 33] may reflect potential ancestral differential effects. While we made every effort to achieve representativeness and increase our sample sizes, we acknowledge that labeling of ancestral populations by self-reported ethnicity does not fully account for genetic variations.

Our results may reflect that ethnicity was self-determined by the participants and potential relationships between them were not investigated. This approach may not always reflect the inter/intra ethnic variation in the frequency distribution of germline variants of the population examined. Also, we were unable to explore additional factors associated with CMPs, including demographic characteristics (socioeconomic status, etc.) and environmental factors (climate conditions, nutritional habits, etc.).

We conclude that, in some populations, the A-3826G, A-1766G, Ala64Thr and A-112C SNPs of *UCP1* gene may be associated with the prevalence of one or more of the most common CMP and their risk factors. Future studies on these SNPs may shed more light on the genetics of CMP and may uncover potential candidates for precision medicine.

## Supporting information

**S1 Checklist. Meta-analysis on genetic association studies checklist.**
(DOCX)

**S1 File. Detailed results and analyses from the case-control study as well as the systematic review and meta-analysis.**
(DOCX)

## Acknowledgments

The authors are grateful to Monika Jasek, Marta Wagner, and Eleftheria Barmpa for their support during the data collection and analysis. We also thank the Center for Precision Genome Editing and Genetic Technologies for Biomedicine, IGB RAS for the provided equipment.

## Author Contributions

**Conceptualization:** Yulii Shidlovskii, Andreas D. Flouris.

**Data curation:** Petros C. Dinas, Eleni Nintou, Maria Vliora, Agata Witkowicz, Anna Kowalska, Paraskevi Gkiata, Yulii Shidlovskii, Andreas D. Flouris.

**Formal analysis:** Petros C. Dinas, Eleni Nintou, Maria Vliora, Anna E. Pravednikova, Paraskevi Sakellariou, Agata Witkowicz, Zaur M. Kachaev, Victor V. Kerchev, Svetlana N. Larina, Lidia Karabon, Yulii Shidlovskii, Andreas D. Flouris.

**Funding acquisition:** Andreas D. Flouris.

**Investigation:** Eleni Nintou, Maria Vliora, Paraskevi Sakellariou, Agata Witkowicz, Zaur M. Kachaev, Victor V. Kerchev, Svetlana N. Larina, James Cotton, Anna Kowalska, Paraskevi Gkiata, Alexandra Bargiota, Zaruhi A. Khachatryan, Anahit A. Hovhannisyan, Mariya A. Antonosyan, Sona Margaryan, Anna Partyka, Pawel Bogdanski, Monika Szulinska, Matylda Kregielska-Narozna, Rafał Czepczyński, Marek Ruchała, Anna Tomkiewicz, Levon Yepiskoposyan, Lidia Karabon, George S. Metsios, Andreas D. Flouris.

**Methodology:** Petros C. Dinas, Maria Vliora, Anna E. Pravednikova, Paraskevi Gkiata, Levon Yepiskoposyan, Lidia Karabon, George S. Metsios, Andreas D. Flouris.

**Project administration:** Andreas D. Flouris.

**Resources:** Andreas D. Flouris.

**Supervision:** Andreas D. Flouris.

**Validation:** Andreas D. Flouris.

**Visualization:** Eleni Nintou, Maria Vliora, Andreas D. Flouris.

**Writing – original draft:** Petros C. Dinas, Eleni Nintou, Maria Vliora, Lidia Karabon, Andreas D. Flouris.

**Writing – review & editing:** Eleni Nintou, Maria Vliora, Anna E. Pravednikova, Paraskevi Sakellariou, Agata Witkowicz, Zaur M. Kachaev, Victor V. Kerchev, Svetlana N. Larina, James Cotton, Anna Kowalska, Alexandra Bargiota, Zaruhi A. Khachatryan, Anahit A. Hovhannisyan, Mariya A. Antonosyan, Sona Margaryan, Anna Partyka, Pawel Bogdanski, Monika Szulinska, Matylda Kregielska-Narozna, Rafał Czepczyński, Marek Ruchała, Anna Tomkiewicz, Levon Yepiskoposyan, Lidia Karabon, George S. Metsios, Andreas D. Flouris.

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
