## [Decision Letter · Decision Letter 0]

8 Nov 2021

PONE-D-21-21760Prevalence of uncoupling protein one genetic polymorphisms and their relationship with cardiovascular and metabolic healthPLOS ONE

Dear Dr. Flouris,

Thank you for submitting your manuscript to PLOS ONE. After careful consideration, we feel that it has merit but does not fully meet PLOS ONE’s publication criteria as it currently stands. Therefore, we invite you to submit a revised version of the manuscript that addresses the points raised during the review process.

We look forward to receiving your revised manuscript.

Kind regards,

Narasimha Reddy Parine, Ph.D

Academic Editor

PLOS ONE

Journal Requirements:

The case-control study was supported by funding from the European Union 7th Framework Program (FP7-PEOPLE-2013-IRSES Grant No. 319010; U-GENE project). The case-control study also received funding by the Russian Foundation for Basic Research (grant 19-34-51003). The Polish research center has received additional funding for the case-control study from the Polish Ministry of Science and Higher Education 2016-2017 international project co-financed W15/7.PR/2016.

NO

NO authors have competing interests

Reviewers' comments:

Reviewer's Responses to Questions

**Comments to the Author**

1. Is the manuscript technically sound, and do the data support the conclusions?

Reviewer #1: Partly

Reviewer #2: Partly

2. Has the statistical analysis been performed appropriately and rigorously? 

Reviewer #1: No

Reviewer #2: N/A

3. Have the authors made all data underlying the findings in their manuscript fully available?

Reviewer #1: No

Reviewer #2: No

4. Is the manuscript presented in an intelligible fashion and written in standard English?

Reviewer #1: No

Reviewer #2: Yes

5. Review Comments to the Author

Reviewer #1: In this manuscript, Dinas and co-workers carried out a population-based genetic epidemiology study, aimed to document the frequency of four UCP1 gene polymorphisms and to examine its association with cardiovascular-metabolic health in 2283 multiethnic European individuals. Besides, the authors have carried out a meticulous meta-analysis cum systemic review to extend their findings. Overall, the manuscript is well written, and the attempt of the authors should be praised. However, there are some significant issues in the manuscript that need to be addressed.

Comments:

1. The authors are advised to revise the language of the manuscript, as some passages are inaccurate or contain semantic and syntactic mistakes. Further, there are several typos and spacing issues in the manuscript (for example: on line 86, the word ‘relat-ed’, on line 154 and 212, the word ‘be-tween’, on line 236, the word ‘statis-tical’, on line 360 ‘SNPs UCP1 SNPs’etc). This needs to be addressed.

2. The most important result of the study might have also been influenced by a selection bias. How were the probands chosen? How do the authors define an individual as Russian, Armenian ethnic, and healthy? This is a population-based genetic epidemiological study, and it is important to briefly describe the influence of ethnicity/race on inter/intra ethnic variation in the frequency distribution of germline variants.

3. I was wondering why there is no data on the clinical and biochemical features of the study subjects. And whether multivariable analysis to evaluate independent associations has been performed. Further, the paper is assessing several associations facing the risks of multiple testing. Correction is warranted.

4. Add a note in the introduction/methods stating the putative function of the studied SNPs (alters expression, alters binding site, etc), chromosome location, and position. Further, the author should comment on the possible molecular mechanism by which the studied polymorphisms influence the disease risk.

5. What is the time span for the study recruitment? Did they validate the genotyping methodology? If yes, please mention it. Also, the authors need to elaborate on whether a phenotype-blind genotyping process was pursued.

6. Several other demographic/environmental factors and social determinants were not explored in this study which could potentially affect the associations. Do the authors think that this study has any limitations? It is missing in the discussion section.

7. What about the association analysis of UCP1 gene polymorphisms with CMP risk, stratified by gender?

8. Potential ancestral differential effects should be discussed more as they present a considerable contribution to the study. Throughout the manuscript, the gene names should be in italics.

Reviewer #2: # General Comments

- The authors performed a genetic association study to test the influence of genetic variants of UCP1 on cardiometabolic pathologies (CMP) and their risk factors. They included Caucasian individuals from different populations. UCP1 is a key protein in the function of brown adipose tissue, which is implicated in the non-shivering thermogenesis. The authors found different allele frequencies between cases (CMP) and controls for some UCP1 SNPs in some populations. In addition, some alleles/genotypes were associated with CMP risk factors. I think this study can be relevant for improving our understanding of the genetic architecture of CMP and the role of UCP1. However, I have some comments and concerns.

# Specific comments

- Line 67: Missing comma before "which".

- Lines 76-78: The G allele of A-3826G associates with higher HDL? This would mean less cardiometabolic risk.

- Line 86: "related" instead of "relat-ed"

- Line 92: do you mean the frequency of the GENOTYPE AG among persons with CMP? Maybe add "AG genotype".

- Lines 102-107: I agree with the point that a better knowledge of the genetic architecture of CMP can help in their prevention and treatment, but this statement focused on UCP proteins is too strong. Cardiometabolic traits are complex, being influenced by many genes with small effect sizes. Improving our knowledge about the impact of UCP1 variants can contribute to precision medicine, but within the context of polygenic scores considering thousand of other variants. It is unlikely that these common variants can have practical relevance in isolation. Please, modify and soften this statement.

- There is any control for population structure? I understand that you performed analyses within each region, removing the influence of genetic clusters across Europe. However, structure could also exist within a population. You had evidence that all individuals in each group belonged to the same race/ethnicity and individuals were not related? If not, please clearly state this as limitation of the study.

- I do not fully understand table 1.1.3 in supplementary. The prevalence is the result of dividing cases with a given genotype by the total sample size. Then, the prevalence should be between 0 and 1. Maybe the prevalence is shown in brackets, while the actual number of genotype carriers is show above? Please, clarify.

- I am not very familiar with the methodology of meta-analyses, this could be also the case for other readers. Could you extend the explanations and rationale of the decisions made? Several choices were made based on previous guidelines. Maybe you could explain briefly the rationale in the supplementary.

- Figure 1: I would avoid the colors in Figure 1. It makes difficult the visualization of the patterns. I think it is enough with the country name at the bottom. I would also add a legend in the plot reminding that patterned bars are "healthy" and solid bars are "CMP".

- I have not seen any control for multiple comparison. When multiple tests are performed, the risk of false positives accumulates across the different P<0.05 tests performed, increasing the probability of type I error. I am aware that this study is using just a few SNPs and they show linkage between them, which decreases the number of independent tests. Under that scenario, a stringent control of false positives like the Bonferroni correction would be too stringent. However, I think the authors could use other approaches less stringent, like the False Discovery Rate. They are considering several populations and phenotypes, so the total number of tests is not low. At minimum, a mention to this aspect should be done in the Discussion, indicating that a control for increased false positive rate has not been done despite the multiple tests performed.

- In line with the previous comment about the risk of false positives, I think it would be good to add some additional evidence about the functional relevance of the variants. The impact of Ala64Thr is clear, but what about variants from non-coding regions? For example, A-112C seems to modify 4 transcription factor binding sites and its region has specific patterns of chromatin accessibility in several tissues (i.e., DNAse). However, a SNP in linkage disequilibrium (rs72941746) alters much more transcription motifs and more proteins binds in that position (from Haploref, see link below). This could be the causal variant behind the associations of A-112C. Indeed, RegulomeD gives a score of 0.81 to this variant for its support as functional, while A-112C gets 0.61 (the score ranges from 0 to 1). I think additional support coming from functional data can be relevant, specially considering the negative results in the meta-analysis.

- https://pubs.broadinstitute.org/mammals/haploreg/haploreg.php

- https://regulomedb.org/regulome-summary/?regions=rs10011540%0D%0Ars72941746&genome=GRCh37&maf=0.01

- line 260: "compared" instead of "com-pared". There are several typos like this through the manuscript. Please, check it.

- Individuals of the Greek population carrying the C allele of A-112C are 37% less likely to develop CMP. However, healthy individuals with the AC genotype have higher BMI than AA carriers for the entire sample. Do you have any potential explanation for this pattern? You mention in Discussion the existence of contrasting patterns between healthy and CMP groups, but I do not see a convincing explanation.

6. PLOS authors have the option to publish the peer review history of their article (what does this mean?). If published, this will include your full peer review and any attached files.

Reviewer #1: No

Reviewer #2: No

---

## [Author Response · Author response to Decision Letter 0]

4 Jan 2022

Please, see submitted document for a point-by-point response to all issues raised by the Reviewers.

---

## [Decision Letter · Decision Letter 1]

14 Feb 2022

PONE-D-21-21760R1Prevalence of uncoupling protein one genetic polymorphisms and their relationship with cardiovascular and metabolic healthPLOS ONE

Dear Dr. Flouris,

Thank you for submitting your manuscript to PLOS ONE. After careful consideration, we feel that it has merit but does not fully meet PLOS ONE’s publication criteria as it currently stands. Therefore, we invite you to submit a revised version of the manuscript that addresses the points raised during the review process.

We look forward to receiving your revised manuscript.

Kind regards,

Narasimha Reddy Parine, Ph.D

Academic Editor

PLOS ONE

Journal Requirements:

Reviewers' comments:

Reviewer's Responses to Questions

**Comments to the Author**

1. If the authors have adequately addressed your comments raised in a previous round of review and you feel that this manuscript is now acceptable for publication, you may indicate that here to bypass the “Comments to the Author” section, enter your conflict of interest statement in the “Confidential to Editor” section, and submit your "Accept" recommendation.

Reviewer #1: All comments have been addressed

Reviewer #2: (No Response)

2. Is the manuscript technically sound, and do the data support the conclusions?

Reviewer #1: Yes

Reviewer #2: (No Response)

3. Has the statistical analysis been performed appropriately and rigorously? 

Reviewer #1: Yes

Reviewer #2: (No Response)

4. Have the authors made all data underlying the findings in their manuscript fully available?

Reviewer #1: No

Reviewer #2: (No Response)

5. Is the manuscript presented in an intelligible fashion and written in standard English?

Reviewer #1: Yes

Reviewer #2: (No Response)

6. Review Comments to the Author

Reviewer #1: In this revised manuscript, the authors have satisfactorily responded to all the concerns from the previous review and made tremendous improvements in the revision. However, a few things need to be taken care of before any publication is warranted.

On line 246-247: the haplotype analysis revealed that CMP individuals 247 were 24% less likely to carry the GAC haplotype, what do you mean by?

On line 400: It should be written as ‘the A-3826G, A-1766G, Ala64Thr and A-112C SNPs of UCP1 gene’

Reviewer #2: # General Comments

- In general, the authors has addressed the comments well. I have just a few specific comments about the revision.

# Specific comments (lines refer to the manuscript version with track changes)

- Line 114: As a modification of my own previous comment, it would be better to say "Improving our knowledge about the impact of UCP1 variants can contribute to precision medicine, within the context of approaches that consider the polygenicity of cardio-metabolic traits (e.g., polygenic risk scores)."

- Line 176: I do not agree with the argumentation made by the authors. I do think that the risk of false positives and false negatives should be considered depending on the question and the goal. Despite this, I respect their decision of not doing it. This has to be stated in the manuscript, as they have already done, so the reader can take this aspect into account when interpreting the results.

- Line 330: Strange wording: "the A-3826G (AG) was the most prevalent of the four SNPs studied in persons with CMP was 40%,". I guess the prevalence was 40% overall. Please, modify.

7. PLOS authors have the option to publish the peer review history of their article (what does this mean?). If published, this will include your full peer review and any attached files.

Reviewer #1: No

Reviewer #2: No

---

## [Author Response · Author response to Decision Letter 1]

16 Feb 2022

Please, see the submitted response letter.

---

## [Editor Report · Decision Letter 2]

21 Mar 2022

Prevalence of uncoupling protein one genetic polymorphisms and their relationship with cardiovascular and metabolic health

PONE-D-21-21760R2

Dear Dr. Flouris,

We’re pleased to inform you that your manuscript has been judged scientifically suitable for publication and will be formally accepted for publication once it meets all outstanding technical requirements.

Kind regards,

Narasimha Reddy Parine, Ph.D

Academic Editor

PLOS ONE

---

## [Editor Report · Acceptance letter]

24 Mar 2022

PONE-D-21-21760R2 

Prevalence of uncoupling protein one genetic polymorphisms and their relationship with cardiovascular and metabolic health 

Dear Dr. Flouris:

I'm pleased to inform you that your manuscript has been deemed suitable for publication in PLOS ONE. Congratulations! Your manuscript is now with our production department. 

Kind regards, 

on behalf of

Dr. Narasimha Reddy Parine 

Academic Editor

PLOS ONE